# The N-Terminal Region of Soybean PM1 Protein Protects Liposomes during Freeze-Thaw

**DOI:** 10.3390/ijms21155552

**Published:** 2020-08-03

**Authors:** Liyi Chen, Yajun Sun, Yun Liu, Yongdong Zou, Jianzi Huang, Yizhi Zheng, Guobao Liu

**Affiliations:** 1Guangdong Provincial Key Laboratory for Plant Epigenetics, College of Life Sciences and Oceanography, Shenzhen University, Shenzhen 518060, China; chenly@tsinghua-sz.org (L.C.); sunyajun43@163.com (Y.S.); sunshine@szu.edu.cn (Y.L.); biohjz@szu.edu.cn (J.H.); yzzheng@szu.edu.cn (Y.Z.); 2The Instrumental Analysis Center of Shenzhen University (Lihu Campus), Shenzhen University, Shenzhen 518060, China; zouyd@szu.edu.cn

**Keywords:** LEA protein, liposome, freeze-thaw, intrinsically disordered, membrane stabilizing protein

## Abstract

Late embryogenesis abundant (LEA) group 1 (LEA_1) proteins are intrinsically disordered proteins (IDPs) that play important roles in protecting plants from abiotic stress. Their protective function, at a molecular level, has not yet been fully elucidated, but several studies suggest their involvement in membrane stabilization under stress conditions. In this paper, the soybean LEA_1 protein PM1 and its truncated forms (PM1-N: N-terminal half; PM1-C: C-terminal half) were tested for the ability to protect liposomes against damage induced by freeze-thaw stress. Turbidity measurement and light microscopy showed that full-length PM1 and PM1-N, but not PM1-C, can prevent freeze-thaw-induced aggregation of POPC (1-palmitoyl-2-oleoyl-sn-glycero-3-phosphocholine) liposomes and native thylakoid membranes, isolated from spinach leaves (*Spinacia oleracea*). Particle size distribution analysis by dynamic light scattering (DLS) further confirmed that PM1 and PM1-N can prevent liposome aggregation during freeze-thaw. Furthermore, PM1 or PM1-N could significantly inhibit membrane fusion of liposomes, but not reduce the leakage of their contents following freezing stress. The results of proteolytic digestion and circular dichroism experiments suggest that PM1 and PM1-N proteins bind mainly on the surface of the POPC liposome. We propose that, through its N-terminal region, PM1 functions as a membrane-stabilizing protein during abiotic stress, and might inhibit membrane fusion and aggregation of vesicles or other endomembrane structures within the plant cell.

## 1. Introduction

Plants have developed various adaptations to maintain normal physiological processes under adverse abiotic stress, such as soil salinity, drought and extreme temperatures. For instance, it has been demonstrated experimentally that late embryogenesis abundant (LEA) proteins are involved in plant tolerance to abiotic stress [1,2,3]. Based on Pfam domains within their sequences, LEA proteins can be classified into eight subgroups [3]. Most research has focused on the protective functions of the LEA_4, LEA_5 and dehydrin subgroups, generating data on expression profile, cellular localization, and the role of the respective genes and proteins. There is relatively little research on the protective roles of other groups, including the LEA_1 proteins, which are highly expressed in mature seeds and also accumulate in vegetative tissues under stress conditions [4,5,6]. Some genetic studies have been performed showing that the over-expression of LEA_1 genes, such as *LEA4-1* from *Brassica napus*, *BhLEA1* and *BhLEA2* from *Boea hygrometrica*, *AtLEA4-5* from *Arabidopsis thaliana*, and *XsLEA1-8* from *Xerophyta schlechteri*, confer tolerance to salt, drought and osmotic stress in transgenic plants [5,6,7]. The over-expression of recombinant *Gastrodia elata GeLEA1-1* or *Xerophyta schlechteri XsLEA1-8* enhances *Escherichia coli* viability under low-temperature or heat stress [7,8]. In addition, in vitro, the recombinant LEA_1 proteins AtLEA4-2 and AtLEA4-5 (both *A. thaliana*) and XsLEA1-8 can preserve lactate dehydrogenase activity and prevent this enzyme aggregating during freeze–thaw, heat, desiccation and oxidative stress [7,9,10]. Liu et al. suggested that BhLEA1 and BhLEA2 play a general protective role in the plant cell during dehydration stress and increase membrane and protein stability, as indicated by the relatively low electrolyte leakage and higher SOD and POD activity in transgenic plant leaves [5]. In contrast, *A. thaliana* basic protein LEA18 (LEA_1) specifically aggregates and destabilizes negatively charged liposomes, which suggests that LEA18 does not function as a membrane-stabilizing protein, but could modulate membrane stability depending on membrane composition [11].

The LEA proteins are a broad family of proteins, many of which are thought to be intrinsically disordered proteins (IDPs) with “moonlighting” activity [3,10]. However, to our knowledge, relatively few individual LEA proteins have been shown to have multifunctional protective activities. One example of a multifunctional protein, PM1, a LEA_1 protein from soybean, can interact with a range of other molecules, such as non-reducing sugars, poly-l-lysine and phospholipids, during dehydration. PM1 is likely to be an important component of the cellular, organic glass that may stabilize desiccation-sensitive proteins and membranes in stressed plants [12]. PM1 contains a high proportion of basic amino acids, including 13 lysine, 5 arginine and 10 histidine residues, but these are distributed unequally along the length of the protein. There are 12 lysine residues and 3 arginine residues located in the N-terminal half (residues 1–84), while 9 histidine residues and 2 arginine residues are located in the C-terminal region (residues 85–173). In previous papers, we have demonstrated that PM1 protein can bind some metal ions (Fe^3+^, Ni^2+^, Cu^2+^ and Zn^2+^) and so play an important protective role in reducing oxidative damage and ion toxicity in plants exposed to abiotic stress [13]. In addition, PM1 or its truncated version, PM1-C (C-terminal half only), but not PM1-N, can form oligomers and high molecular weight (HMW) complexes via its C-terminal histidine residues both in vitro and *in planta*. Crucially, binding of Cu^2+^ at high concentrations, which takes place through the same histidine residues, seems to promote oligomerization and the formation of HMW complexes by PM1 [14].

In the present paper, we investigate whether soybean LEA_1 protein PM1 and its two truncated forms, PM1-N and PM1-C, possess the ability to suppress freeze-thaw-induced damage of liposomes. We show that only PM1 and PM1-N can effectively prevent aggregation of POPC liposomes caused by freeze-thaw treatment. Furthermore, we show that, although PM1 and PM1-N inhibit membrane fusion events, they do not reduce leakage of POPC liposomes. On the basis of these observations, we propose that the N-terminal region of PM1 protein has a membrane stabilizing function during abiotic stress.

## 2. Results

### 2.1. PM1 and PM1-N Proteins Inhibit the Freeze-Thaw-Induced Increase in Turbidity of a Liposome Suspension

The cell membrane is composed of lipids (phospholipids and cholesterol), proteins and carbohydrate groups. Among the phospholipids, phosphatidylcholine (PC) represents a major component of biological membranes and the PC, POPC (1-palmitoyl-2-oleoyl-sn-glycero-3-phosphocholine), can be used to produce liposomes that model cell membranes. Light scattering due to aggregation of POPC liposomes can be measured by apparent absorbance at 400 nm [15]. The OD_400_ of a freshly prepared POPC liposome suspension was ~0.3 and the addition of one of the proteins thaumatin, PM1, PM1-N or PM1-C to POPC liposomes did not obviously affect their turbidity before exposure to stress (Figure 1).

The turbidity, as shown by OD_400_, of the liposome suspension increased markedly to 0.8 after three freeze-thaw cycles, indicating that the freeze-thaw treatment causes liposome aggregation. When 0.8 mg/mL thaumatin (molar ratio of protein:lipid~1:3300) or 0.8 mg/mL PM1-C protein (molar ratio of protein:lipid~1:1650) was added to the POPC liposome suspension, its OD_400_ increased to 0.6 after the freeze-thaw treatment, which is significantly different (*p* < 0.01) to that of liposomes only. Thaumatin was chosen as a negative control in the following study because of having no known function in membrane protection [11] and its similar molecular size to PM1 protein. There was no statistical difference in turbidity between liposome/PM1-C and liposome/thaumatin. However, when PM1 or PM1-N protein (0.04–0.4 mg/mL) was included in the liposome suspension, the turbidity increase after freeze-thaw was appreciably suppressed and the extent of this suppression was dependent on the concentration of the added protein, being significantly (*p* < 0.05) or extremely significantly different (*p* < 0.01) from that of thaumatin. When the concentration of PM1 (molar ratio of protein:lipid ~1:3300) or PM1-N protein (molar ratio of protein:lipid ~1:1650) reached 0.8 mg/mL, the post-treatment turbidity was 0.4 at OD_400_, i.e., similar to that before freeze-thaw.

### 2.2. PM1 and PM1-N Proteins Can Inhibit or Prevent Freeze-Thaw-Induced Aggregation of Liposomes and Isolated Thylakoid membranes

The inhibitory effect of PM1, PM1-N or PM1-C protein on liposome aggregation can be followed by light microscopy. Before freeze-thaw treatment, POPC liposomes were small in size and evenly distributed in the suspension (Figure 2A). The addition of one of thaumatin, PM1, PM1-N or PM1-C proteins to POPC liposomes did not cause aggregation before freeze-thaw treatment, but in the absence of added proteins, liposome aggregation could be seen to occur after freeze-thaw treatment. With either 0.8 mg/mL thaumatin (negative control) or 0.8 mg/mL PM1-C protein in the suspension, the liposomes still aggregated after freeze-thaw. In contrast, when 0.8 mg/mL PM1 or 0.8 mg/mL PM1-N protein was added to liposomes, the suspension remained homogeneous even after freeze-thaw.

Next, the anti-aggregation effect of the PM1, PM1-N or PM1-C proteins on natural biological material was tested using thylakoid membranes isolated from spinach leaves (*Spinacia oleracea*). Under the light microscope, the thylakoid membranes appeared as evenly distributed, small particles in suspension (Figure 2B). The addition of thaumatin, PM1, PM1-N or PM1-C proteins individually to thylakoid membranes did not cause aggregation. However, after freeze-thaw treatment, the thylakoid membranes aggregated markedly, as was apparent by light microscopy, and even being directly visible to the naked eye. The addition of 0.8 mg/mL thaumatin (negative control) or 0.8 mg/mL PM1-C protein did not prevent the formation of thylakoid aggregates when the suspension was subjected to freeze-thaw. In contrast, when 0.8 mg/mL PM1 or 0.8 mg/mL PM1-N protein was added to the thylakoid suspension, there was almost no aggregation after freeze-thaw.

### 2.3. The Addition of Either PM1 or PM1-N Protein Can Stabilize the Liposome Particle Size

The particle size distribution of POPC liposomes was measured and a single peak was obtained by dynamic light scattering (DLS) centering around 130 nm before freeze-thaw treatment (Figure 3). The addition of thaumatin, PM1, PM1-N or PM1-C individually to liposomes did not affect the particle size distribution (Figure 3A). After freeze-thaw treatment, another peak of 800 nm was observed besides the main peak of 130 nm, indicating that freeze-thaw can cause liposome adhesion, involving liposome fusion and/or aggregation. When either 0.8 mg/mL thaumatin or 0.8 mg/mL PM1-C protein was added to the liposome suspension, a peak at 250–270 nm was present besides the main peak of 130 nm. In contrast, when either 0.8 mg/mL PM1 or 0.8 mg/mL PM1-N protein was added to the liposome suspension, a small peak of particle size ~50 nm emerged besides the main peak of 140 nm (Figure 3B).

Together, the above results suggest that PM1 and PM1-N, but not PM1-C, can inhibit aggregation of liposomes and thylakoids caused by freeze-thaw treatment; the effect was concentration-dependent within a certain range and also specific. Furthermore, PM1 and PM1-N protein can prevent membrane fusion in POPC liposomes due to freeze-thaw.

### 2.4. PM1 and PM1-N Proteins Cannot Prevent Freeze-Thaw-Induced Leakage of Liposomes

Freeze-thaw treatment can cause the collapse of liposome membranes, causing leakage of liposome contents. To obtain a deeper insight into the protective effects of PM1 protein and its N-terminal region, a leakage experiment, using the fluorescent probe, carboxyfluorescein (CF), trapped within liposomes, was carried out according to [16]. As shown in Figure 4, the leakage rate of pure POPC liposomes after freeze-thaw treatment was 91.9%, suggesting that freeze-thaw almost entirely destroys their integrity. In the presence of the proteins (all at 0.8 mg/mL) used in this study, the post-treatment leakage rates of POPC liposomes were as follows: thaumatin, 86.8%; PM1, 80.1%; PM1-N, 80.5%; PM1-C, 79.6%. There was no statistical difference in CF leakage rates between any of the liposome/protein combinations and the liposome-only samples, suggesting that PM1 and its truncated versions prevent leakage no better than thaumatin or, indeed, no protein (Figure 4).

### 2.5. PM1 and PM1-N Proteins Can Effectively Inhibit or Prevent Membrane Fusion

The leakage of soluble contents from liposomes is often accompanied by membrane fusion [17]. The degree of fusion of the membrane can be assessed quantitatively by fluorescence resonance energy transfer (FRET) [18]. In the present paper, POPC liposomes whose membranes contained a pair of fluorescently labeled phospholipids (Rh-PE and NBD-PE) that can undergo FRET were mixed with unlabeled liposomes. When membrane fusion between labeled and unlabeled liposomes occurs, the degree of FRET between Rh-PE and NBD-PE will be reduced, due to dilution of the fluorescent probes. As shown as Figure 5, about 41.6% of liposomes underwent membrane fusion after freeze-thaw. When 0.8 mg/mL thaumatin was added to the liposome suspension, the degree of membrane fusion was 37.1% after freeze-thaw is not significantly different from that of liposomes only. When 0.8 mg/mL PM1-C protein was added to the liposome suspension, the degree of membrane fusion was 31.7% after freeze-thaw, showing a significant difference (*p* < 0.05) with liposomes only or thaumatin as negative control. However, in the presence of 0.8 mg/mL PM1 or 0.8 mg/mL PM1-N, the degree of membrane fusion reduced markedly to 10.6% or 8.6%, respectively, showing significant differences (*p* < 0.01) to liposomes only or thaumatin as negative control. Thus, the ability of PM1 to reduce membrane fusion in POPC liposomes caused by freeze-thaw largely resides in its N-terminal sequence.

The above results show that PM1 or PM1-N protein do not reduce the leakage of CF from POPC liposomes subjected to freeze-thaw, although they can significantly inhibit membrane fusion. Compared to thaumatin protein as negative control, on the other hand, PM1-C protein did not reduce leakage, and only reduced membrane fusion of POPC liposomes to some extent. 

### 2.6. The Presence of POPC Liposomes Does Not Affect Digestion of PM1 by Trypsin

PM1 is an IDP, a group of proteins that are on the whole degraded ~100 times faster than folded proteins [19]. Liposomes can increase the folding of IDPs, and this can be measured by a limited proteolysis test [20,21]. Trypsin is a Lys-specific protease that can recognize the Lys residues located exclusively within the N-terminal half of the soybean PM1 sequence. Besides the main band at ~20 kDa, the purified PM1 protein contains several smaller bands, which also can be detected by PM1 specific antibody [14], probably indicating a small amount of degradation in protein preparation. Figure 6A shows that PM1 was readily and gradually degraded by trypsin over 30 min, as indicated by the main band at ~20 kDa becoming weaker and by the appearance of smaller bands below 17 kDa. This pattern of degradation did not change in the presence of POPC liposomes, suggesting that the structure of PM1 protein remains disordered state and is not influenced by the presence of liposomes.

### 2.7. The Presence of POPC Liposomes Does Not Change the Secondary Structure of PM1 Protein

It has been reported that *Zea mays* dehydrin DHN1 and *A. thaliana* dehydrin Lti30 bind the negatively charged head groups of phospholipids and that this is accompanied by an increase in α-helicity of both LEA proteins [15,22,23,24]. We tested the effect of POPC liposomes on PM1 secondary structure using circular dichroism (CD) (Figure 6B). We observed that PM1 protein remained unstructured both before, and after, freeze-thaw in the absence or presence of liposomes, as indicated by an ellipticity minimum at 198 nm. 

Together, the results of trypsin digestion and CD analysis suggest that PM1 interacts mainly with the surface of the uncharged POPC liposomes.

## 3. Discussion

LEA proteins, a large and highly diverse protein family, accumulate during seed desiccation in the later stages of embryogenesis. Given the diversity and compartmentalization of LEA proteins in plant tissues, a detailed understanding of their function involves characterization of each protein individually. The over-expression of some LEA protein genes in transgenic plants confers tolerance to low-temperature stress. For example, over-expression of *A. thaliana COR15A* and *COR15B* (LEA_4) in Arabidopsis or of *Citrus unshiu CuCOR19* (dehydrin) in tobacco (*Nicotiana tabacum*) increases the freezing tolerance of the resulting transgenic plants, as indicated by reduced electrolyte leakage compared to controls [25,26,27]. In vitro, COR15A and COR15B stabilize liposomes, either in terms of freeze-induced solute leakage or membrane fusion [26]. Similarly, *Pisum sativum* LEAM (LEA_4) or the *Artemia franciscana* AfrLEA2 and AfrLEA3m (LEA_4) increase liposome stability after freeze-thawing, as illustrated by a reduced leakage of entrapped CF molecules [28,29]. In contrast, three *A. thaliana* LEA proteins, LEA1, LEA26 and LEA27 (LEA_2), increase CF leakage from liposomes after freeze-thaw treatment [16]. Arabidopsis Lti30 (dehydrin) assembles lipid vesicles and native thylakoid membranes into aggregates by a possible role in membrane cross-linking [15], while Arabidopsis LEA18 protein (LEA_1) increases CF leakage, liposome membrane fusion and liposome aggregation under non-freezing conditions [11]. 

Shih et al. reported that soybean PM1 protein can interact with POPC lipids to maintain the liquid-crystal phase over a wide temperature range in the dry state [12]. In the present paper, we found that at high concentrations PM1/PM1-N protein specifically binds to or interacts with the surface of liposome membranes in solution and in the frozen state. For this reason, we speculate that, in a solution of PM1 or PM1-N, the liposomes are kept separate from each other by the protein molecules. During freeze-thaw, liposomes break down or fragment such that their contents (CF in our experiments) leak out [26]. In the absence of PM1 and PM1-N proteins, the resulting membrane fragments can fuse and POPC liposomes aggregate to such an extent that they become visible under the light microscope and even to the naked eye. In the presence of PM1 and PM1-N, the adherence of liposome membranes was reduced and membrane fusion and liposome aggregation was reduced, even though leakage of CF contents was not prevented. Therefore, our observations support a protection mechanism for PM1 on cells and cell membranes that is very similar to the “molecular shield” model. This model was proposed by Wise and Tunnacliffe to explain the protective function of LEA proteins towards other proteins under water stress conditions [30].

Some dehydrins, such as *Zea mays* DHN1, *Thellungiella salsuginea* TsDHN1 and TsDHN2, and *A. thaliana* ERD14 and Lti30, bind phospholipids in vitro [15,19,22,23,31]. These interactions between dehydrins and phospholipid molecules are driven by electrostatics and depend on the positively charged residues in the characteristic lysine-rich K-segment of the dehydrins, which pair with the negatively charged head groups of the lipids [24]. Ananlysis of the amino acid composition of soybean basic PM1 protein shows that the N-terminal is also rich in lysine residues and has +6 positively charged amino acids in solution at pH 7.4. It is reasonable to speculate that the N-terminal region of PM1 might interact with the surface of liposomes, especially with the negatively charged head groups of phospholipid molecules. 

On the other hand, the C-terminal region of PM1 contains 9 histidine residues and has only +1 positively charged amino acids at pH 7.4. In contrast to the PM1 and PM1-N proteins, PM1-C cannot effectively prevent liposome aggregation and membrane fusion during freeze-thaw. This could be because PM1-C, with many fewer charged residues, can only attach to the liposome surface with low affinity. Therefore, it is not as effective at keeping liposomes separate and consequently preventing them from aggregating. Therefore, the anti-aggregation function of PM1 protein on cell membranes is likely mainly conferred by its N-terminal region.

*A. thaliana* basic LEA18 protein and soybean PM1 protein both belong to the LEA_1 subgroup of LEA proteins. However, there is only 30% similarity between the two protein sequences. Intriguingly, LEA18 does not function as a membrane-stabilizing protein, in contrast to PM1. Instead, the LEA18 protein causes aggregation and destabilization of negatively charged liposomes [11]. 

We have carried out a series of studies on the protective functions of soybean PM1 protein. The amino acid sequence shows that its histidine residues are mainly located in the C-terminal half, while its lysine residues are mostly in the N-terminal region. The C-terminal region has the potential to bind metal ions, scavenge hydroxyl radicals, and form oligomers and HMW complexes. These functions are directly related to the number of histidine residues in the C-terminal region [13,14]. In contrast to the disordered nature of the C-terminus, the highly conserved N-terminal half of PM1 can be induced to form significant levels of α-helical structure by SDS or trifluoroethanol [14]. At the same time, this N-terminal region of PM1 has a protective effect on LDH activity (Figure A1) and can also effectively prevent liposome aggregation. Both these functions may relate to the number of lysine residues in the N-terminal region. It has been speculated that the two halves of PM1 could perform different functions by binding different targets. The conformational flexibility and structural plasticity of PM1, as an IDP, may be the molecular basis for its multiple protective functions.

## 4. Materials and Methods 

### 4.1. Protein Over-Expression and Purification

Immature soybean *(G. max* L. Merr. cv Bainong 6#) seeds were collected from pods 35–45 d after flowering and then total RNA was extracted from plant material with Trizol reagent (TAKARA, Otsu, Japan). The full-length ORF of the *PM1* gene was amplified by RT-PCR using the PrimeScript^TM^ one-step RT-PCR kit (TAKARA, Otsu, Japan). Constructs containing the full-length soybean *PM1*, *PM1-N* and *PM1-C* sequences were described previously [13,14]. Recombinant PM1, PM1-N and PM1-C proteins were purified using affinity chromatography, and His-tags were removed by incubation with thrombin as previously described [14]. The proteins were lyophilized and then stored at −80°C for later use. The proteins were resuspended in a suitable buffer before use. 

### 4.2. Preparation of Liposomes

POPC (1-palmitoyl-2-oleoyl-sn-glycero-3-phosphocholine) was obtained from Avanti Polar Lipids (Alabaster, AL, USA). Large unilaminar vesicles (LUVs, 100 nm) of POPC were prepared by the extrusion method described previously [15]. Briefly, the lipids were dissolved in chloroform, and lipid mixtures were dried under a gentle liquid nitrogen flow and subsequently rehydrated in 10 mM Na_2_HPO_4_-KH_2_PO_4_ buffer, pH7.4. The lipid solution was extruded in an Avanti Mini Extruder (100 nm polycarbonate filter), repeated 15 times and then liposomes were collected. 

For the leakage experiment, liposomes were made according to [16]. Briefly, an appropriate amount of POPC lipid was hydrated in buffer (250 μL 100 mM CF (carboxyfluorescein) in 10 mM Na_2_HPO_4_-KH_2_PO_4_ buffer, pH 7.4). The lipid solution was extruded as above and liposomes were collected. To remove external CF, the lipid samples were passed through a Sephadex G-25 column (NAP-5, GE Healthcare, Milwaukee, WI, USA) in buffer (10 mM Na_2_HPO_4_-KH_2_PO_4_, 0.1 mM EDTA and 50 mM NaCl). 

To measure membrane fusion, two liposome samples were prepared: One containing 1 mol % of both NBD-PE and Rh-PE, while the other contained only unlabeled lipids. After extrusion, the two samples were mixed at a 1:9 (labeled:unlabeled) ratio. All liposomes were diluted to 20 mg/mL before use.

### 4.3. Thylakoid Membrane Preparation

Thylakoids were isolated from spinach (*Spinacia oleracea*) as previously described [32]. Briefly, 40 g spinach leaves and 100 mL cold A buffer (0.3 M Suc, 50 mM Na-phosphate, pH 7.4, and 5 mM MgCl_2_) was mixed for 5 × 10 s with a coldmixer. The solution was filtered with 50 μM nylon mesh and centrifuged at 3000 rpm for 3 min. The pellet was suspended in 30 mL A buffer and centrifuged at 4500 rpm for 5 min. The solution was homogenized in 30 mL B buffer (10 mM phosphate, pH 7.4, 5 mM MgCl_2_, and 5 mM NaCl) and centrifuged at 4500 rpm for 5 min. The pellet (about 200 mg) was homogenized in 10 mL buffer (0.1 M Suc, 10 mM phosphate, pH 7.4, 5 mM MgCl_2_ and 5 mM NaCl).

### 4.4. Freeze-Thaw Treatment of Liposomes

Either liposomes (20 mg/mL) or thylakoid membranes (about 20 mg/mL) were mixed with the same volume of the PM1, PM1-N and PM1-C proteins at concentrations of 0.08, 0.2, 0.8 or 1.6 mg/mL or the negative control protein thaumatin (from *Thaumatococcus daniellii*; Sigma, St. Louis, MO, USA) at a concentration of 1.6 mg/mL in 10 mM Na_2_HPO_4_-KH_2_PO_4_ buffer, pH 7.4, or with buffer alone. The freeze-thaw treatment was performed as previously described [26]; samples were then frozen in an ethylene glycol bath at −20 °C for 2 h and allowed to thaw for 30 min at 28°C. This freeze-thaw cycle was repeated up to three times. 

### 4.5. Turbidity Measurement and Dynamic Light Scattering (DLS) 

A simple aggregation assay was performed by measuring apparent absorbance [33], namely turbidity, due to light scattering at 400 nm (denoted as OD_400_), with an Ultro Spec 2000 (GE Healthcare, Milwaukee, WI, USA). In addition, particle size distribution in suspension was measured using a DLS analyzer (Zetaplus Zeta Potential Analyzer, Brookhaven Instruments, Holtsville, NY, USA) [11]. For both of these measurements, the samples were diluted to avoid saturation. 

### 4.6. Light Microscopy 

For light microscopy, 6 μL samples were placed on a glass slide, and images were examined using the optical microscope Olympus BX51 (Olympus Corporation, Tokyo, Japan).

### 4.7. CF Leakage Experiments

The leakage experiments were performed as previously reported [16]. Same of 12 μL were diluted with 300 μL Na_2_HPO_4_-KH_2_PO_4_ buffer in 96-well plates. CF fluorescence was measured with a Varioskan Flash multimode reader (Thermo Scientific, Waltham, MA, USA) with the following settings: Excitation 444 nm; emission 555 nm. While fluorescence is strongly quenched at the high concentration inside intact liposomes, fluorescence will increase when CF is released into the surrounding buffer. The 100% fluorescence level for leakage (i.e., complete release of entrapped CF) was obtained by detergent lysis of the liposomes with 5 μL 0.1% Triton X-100 solution.

### 4.8. Membrane Fusion Measurements

Membrane fusion was measured as a reduction of fluorescence resonance energy transfer (FRET) between Rh-PE and NBD-PE by measuring the increase in NBD fluorescence, due to the dilution of the probes after fusion of labeled, and unlabeled, liposome samples [18]. This is using a Hitachi F-4500 fluorescence instrument (Hitachi, Tokyo, Japan) at an excitation wavelength of 450 nm and an emission wavelength of 530 nm. The 100% fusion level (i.e., maximal NBD fluorescence in each sample) was determined after lysis of the liposomes with Triton X-100 solution.

### 4.9. Limited Proteolysis 

Protease sensitivity was tested with trypsin [15]. Typically, PM1 (1 mg/mL) was mixed with the same volume of liposomes (1 mg/mL). To start the digestion, trypsin was added at a mass ratio of 1:200, trypsin to PM1. After the samples were incubated at room temperature for 0–30 min, the reactions were terminated by adding 1 mM phenylmethanesulfonyl fluorid, a trypsin inhibitor. All samples were mixed with 5×loading buffer, heated at 95 °C for 5 min and then run on a 15% ready-made SDS-PAGE gel. 

### 4.10. Far UV-Circular Dichroism Spectroscopy

PM1 protein (20 μM) was mixed with the same volume of liposomes (2 mg/mL) or with pure buffer with or without freeze-thaw treatment. Far UV-circular dichroism (CD) spectra were recorded using a Jasco J-815 CD spectropolarimeter (JASCO Analytical Instruments, Tokyo, Japan). The acquisition parameters were 0.5 nm resolution, 1.0 nm bandwidth, 0.5 s response and 250–190 nm wavelength range. Three spectra were averaged and smoothed to reduce noise.

### 4.11. In Vitro Lactate Dehydrogenase Assays

Lactate dehydrogenase (LDH) assays were adapted from [34]. In short, LDH from rabbit muscle (Roche, Mannheim, Germany) was diluted in 100 mM sodium phosphate buffer (pH 7.0) to a final concentration of 0.357 μM. PM1, PM1-N and PM1-C proteins or the negative control protein lysozyme were added to equal volumes of LDH at molar ratios of 1:1 (test protein: LDH). The enzyme with and without added proteins was frozen in an ethylene glycol bath at −20 °C for 2 h and allowed to thaw for 30 min at 28 °C. This freeze-thaw cycle was repeated up to five times. The activity of LDH was assayed according to [34].

### 4.12. Statistics

*p*-Value was determined using SPSS 9 software (SPSS, Inc., Chicago, IL, USA). One-way ANOVA and Tukey post hoc test using InStat3 (GraphPad Software, San Diego, CA, USA) were performed for all the experiments. All experiments were performed at least in triplicate.

## Figures and Tables

**Figure 1 ijms-21-05552-f001:**
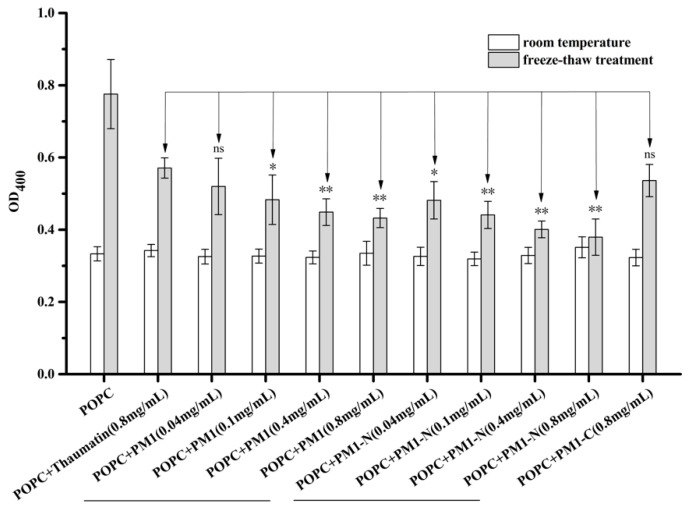
PM1 and its N-terminal region prevent freeze-thaw-induced liposome aggregation. Turbidity of POPC liposomes before and after freeze-thaw treatment in the presence and absence of PM1, PM1-N or PM1-C. For the freeze-thaw test, the samples were frozen in an ethylene glycol bath at −20 °C for 2 h and allowed to thaw for 30 min at 28 °C. This freeze-thaw cycle was repeated up to three times. ** denotes significance at *p* < 0.01, * denotes *p* < 0.05 and “ns” denotes not significant using one-way ANOVA, plus a Tukey post-hoc test.

**Figure 2 ijms-21-05552-f002:**
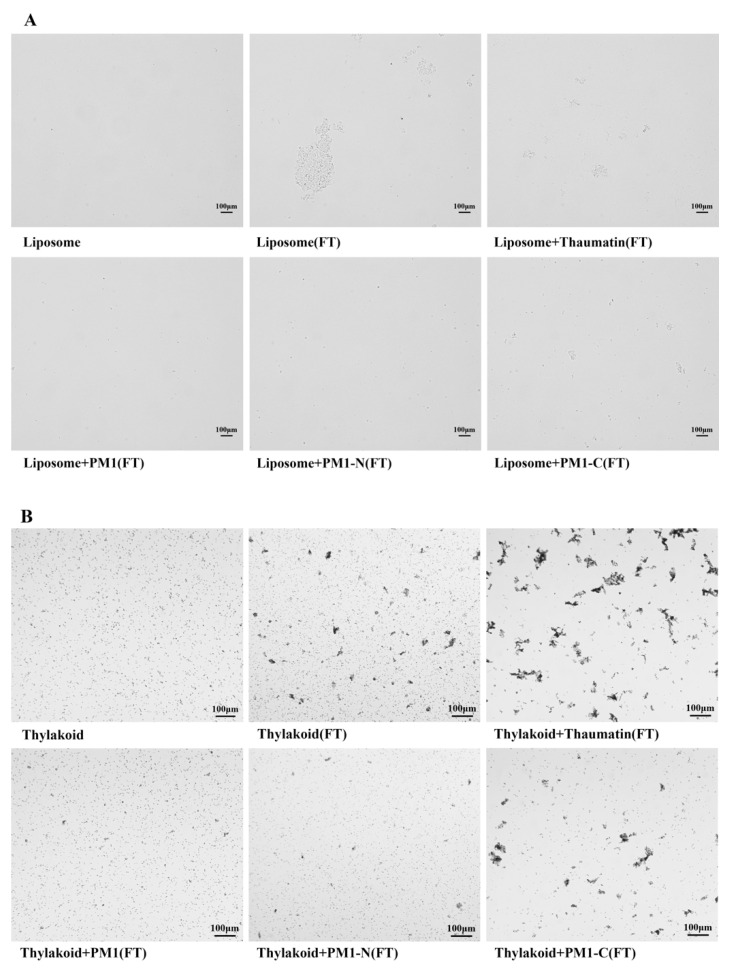
PM1 and its N-terminal region prevent POPC liposome or thylakoid aggregation resulting from freeze-thaw treatment as assessed by light microscopy. (**A**) POPC liposomes; (**B**) spinach leaf thylakoids. Bars in (**A**) and (**B**) represent 100 μm. FT: freeze-thaw treatment.

**Figure 3 ijms-21-05552-f003:**
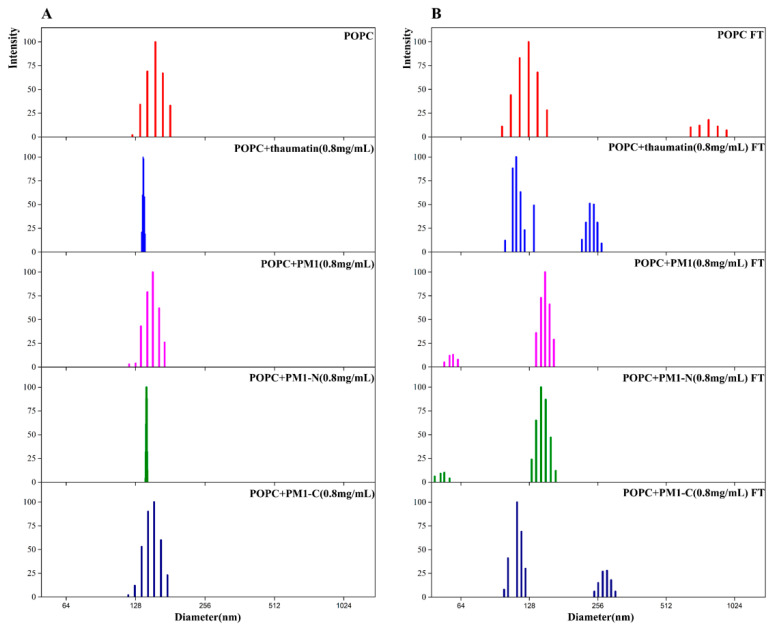
Size distribution of POPC liposomes before and after freeze-thaw treatment in the presence and absence of PM1, PM1-N or PM1-C. (**A**) Size distribution of POPC liposomes before treatment; (**B**) size distribution of POPC liposomes after freeze-thaw treatment.

**Figure 4 ijms-21-05552-f004:**
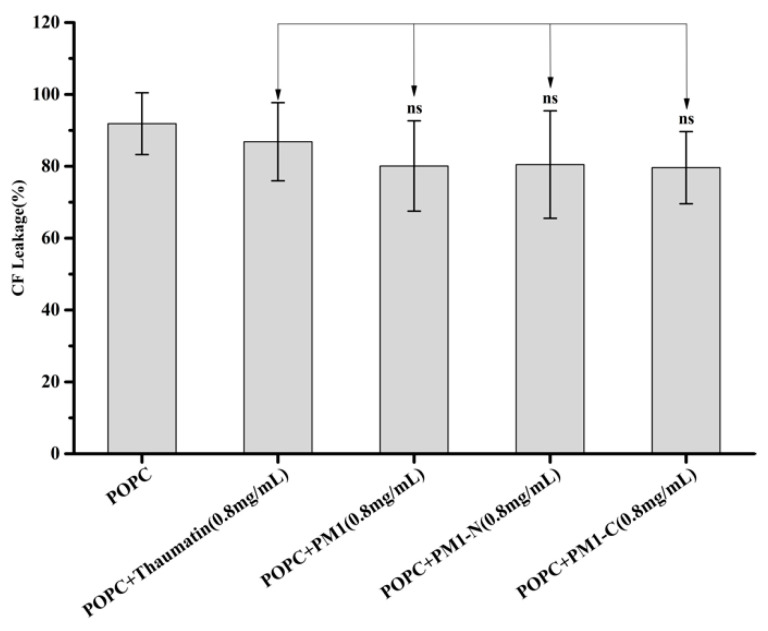
Influence of PM1, PM1-N and PM1-C on CF leakage from liposomes after freeze-thaw treatment. POPC liposomes were subjected to freeze-thaw in the absence or presence of PM1, PM1-N or PM1-C, or thaumatin as a negative control protein. “ns” denotes not significant using one-way ANOVA, plus a Tukey post-hoc test.

**Figure 5 ijms-21-05552-f005:**
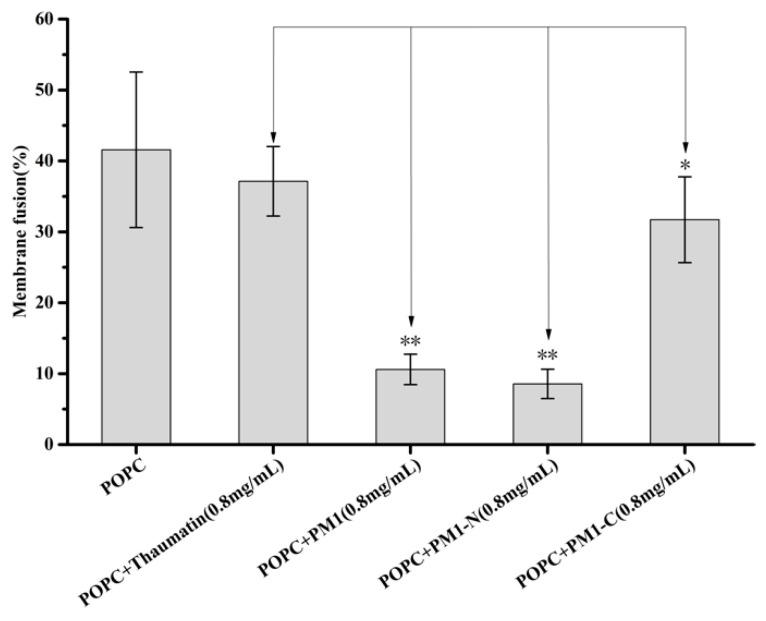
Effect of PM1, PM1-N and PM1-C on membrane fusion caused by freeze-thaw treatment. POPC liposomes were subjected to freeze-thaw in the absence or presence of PM1, PM1-N or PM1-C, or thaumatin as a negative control protein. Membrane fusion was determined by a FRET assay. ** denotes significance at *p* < 0.01and * denotes *p* < 0.05 using one-way ANOVA, plus a Tukey post-hoc test.

**Figure 6 ijms-21-05552-f006:**
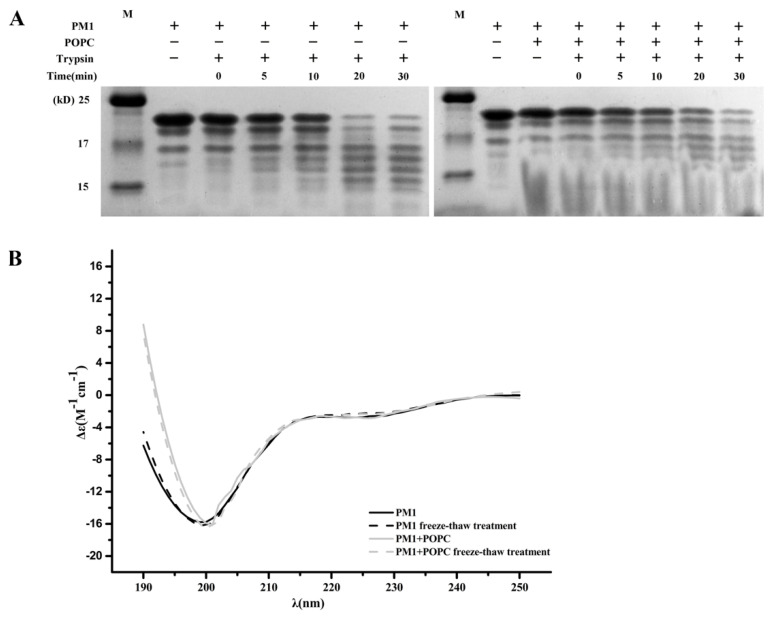
PM1 does not gain structure in the presence of liposomes. (**A**) Digestion of PM1 by trypsin in the presence or absence of POPC liposomes. (**B**) CD spectra of PM1 alone and PM1 in the presence of liposomes before and after freeze-thaw treatment.

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
