# Peer review of "The N-Terminal Region of Soybean PM1 Protein Protects Liposomes during Freeze-Thaw"

_ijms, 2020, doi:10.3390/ijms21155552_

Round 1
Reviewer 1 Report
Chen et al present an investigation of the late embryogenesis abundant group 1 protein PM1 from soybeans. They show that full-length PM1 and it's N-terminal half can protect liposomes against damage induced by freeze-thaw stress. They show this through turbidity measurements, microscopic observation and dynamic light scattering measurements of liposome aggregation. They then go on to show that PM1 and it's N-terminal half (PM1-N) can inhibit liposome membrane fusion through a FRET-based liposome fusion assay, but that these proteins do not stop liposome leakage following freezing. The finally show that PM1 and PM1-N are associated with the surface of liposomes and propose that they act as membrane-stabilising proteins.
Overall the paper is succinctly written and generally well presented. There are a few grammatical errors here and there, but not so many that the meaning is not understandable. I have only minor suggestions for the authors.
Minor issues:
- L93 - What is the purpose of using thaumatin? It is mentioned in the methods section that this is a negative control, but it's utilisation as a negative control is not explained in the text until L100. The reason for using thaumatin as a negative control should be stated at the time it is first mentioned.
- Figure 2. The microscopy images themselves are fine, but I think this section could benefit from some objective measurement of the aggregation seen under the microscope (i.e. some way of quantitating aggregation from the images taken).
- Figure 3. These 3D view of these charts makes them difficult to read and the data points hard to interpret. I feel this figure would be better presented as a series of standard bar charts or in some other 2D format that is easier to read and interpret.
- L200 - I think the word "although" is missing from between "freeze-thaw" and "they can significantly…".
- Figure 6. The purified PM1 protein contains several other bands - this is not mentioned in the text. Do the authors know the identities of these extra bands? Are they degradation products? Is there some explanation for these extra bands? While it appears clear that PM1 can be digested by trypsin and these extra bands do not necessarily interfere with this result, it would be good to at least have some discussion of the purity of the PM1 preparation in the manuscript and acknowledgement of these extra bands.
- There does not appear to be adequate support for the statement in the abstract that PM1 and PM1-N "do not insert into the membrane". From the experiments performed it cannot be ruled out that PM1 partially inserts into the membrane. I suggest the authors add some detail surrounding this point to the discussion and remove this phrase from the abstract. Or, alternatively, perform the experiments that would be required to show that there is no membrane insertion of PM1.
Author Response
Response to Reviewer 1 Comments
Point 1: L93 - What is the purpose of using thaumatin? It is mentioned in the methods section that this is a negative control, but it's utilisation as a negative control is not explained in the text until L100. The reason for using thaumatin as a negative control should be stated at the time it is first mentioned.
Response 1: According to the suggestion, we have added the explanation in L99-101.
Point 2: Figure 2. The microscopy images themselves are fine, but I think this section could benefit from some objective measurement of the aggregation seen under the microscope (i.e. some way of quantitating aggregation from the images taken).
Response 2: It is difficult to quantitatively analyze aggregation from the microscopy images,although the aggregation in tubes can even be directly visible to the naked eye. Therefore, we have quantitated aggregation by measuring turbidity instead, as shown in the manuscript.
Point 3: Figure 3. These 3D view of these charts makes them difficult to read and the data points hard to interpret. I feel this figure would be better presented as a series of standard bar charts or in some other 2D format that is easier to read and interpret.
Response 3: As the reviewer requests, we have now changed Figure 3 from 3D view to 2D format.
Point 4: L200 - I think the word "although" is missing from between "freeze-thaw" and "they can significantly…".
Response 4: Thank you, we have corrected this error. Please see L201.
Point 5: Figure 6. The purified PM1 protein contains several other bands - this is not mentioned in the text. Do the authors know the identities of these extra bands? Are they degradation products? Is there some explanation for these extra bands? While it appears clear that PM1 can be digested by trypsin and these extra bands do not necessarily interfere with this result, it would be good to at least have some discussion of the purity of the PM1 preparation in the manuscript and acknowledgement of these extra bands.
Response 5: These extra bands are degradation products of PM1 protein, which have been found in our previous study. We have added the explanation in L208-210.
Point 6: There does not appear to be adequate support for the statement in the abstract that PM1 and PM1-N "do not insert into the membrane". From the experiments performed it cannot be ruled out that PM1 partially inserts into the membrane. I suggest the authors add some detail surrounding this point to the discussion and remove this phrase from the abstract. Or, alternatively, perform the experiments that would be required to show that there is no membrane insertion of PM1.
Response 6: We agree and have now removed this phrase from the Abstract and Results section. Please see L27-29 and L227-228.
Reviewer 2 Report
The authors reported an interesting study regarding the N-terminal region of the soybean PM1 protein protects liposomes during freeze-thaw.
The manuscript is well written and easy to read, fitting the scope of the Journal.
I have just some consideration on the M&M section.
The authors should report some information regarding the material investigated (soybean), for example the name of the cultivar, how the materials were produced etc. Only the reference is not sufficient for this type of work. Several agronomic and environmental conditions can affect the crop growth and its physiological behaviour.
Author Response
Response to Reviewer 2 Comments
Point : I have just some consideration on the M&M section.The authors should report some information regarding the material investigated (soybean), for example the name of the cultivar, how the materials were produced etc. Only the reference is not sufficient for this type of work. Several agronomic and environmental conditions can affect the crop growth and its physiological behaviour.
Response : We have added this further information in the Materials and Methods section. Please see L295-299.